# Fetal Hemoglobin as a Predictive Biomarker for Retinopathy of Prematurity: A Prospective Multicenter Cohort Study in Portugal

**DOI:** 10.3390/biomedicines13010110

**Published:** 2025-01-06

**Authors:** Mariza Fevereiro-Martins, Laura Aguiar, Ângela Inácio, Carlos Cardoso, Ana Carolina Santos, Carlos Marques-Neves, Hercília Guimarães, Rui Pinto, Manuel Bicho

**Affiliations:** 1Ecogenetics and Human Health Unit, Environmental Health Institute (ISAMB), Associate Laboratory TERRA, Faculty of Medicine, University of Lisbon, Av. Professor Egas Moniz, 1649-028 Lisbon, Portugal; lauramsaguiar@hotmail.com (L.A.); minacio@medicina.ulisboa.pt (Â.I.); carolinasantos@medicina.ulisboa.pt (A.C.S.); cmneves@medicina.ulisboa.pt (C.M.-N.); manuelbicho@medicina.ulisboa.pt (M.B.); 2Institute for Scientific Research Bento da Rocha Cabral, Calçada Bento da Rocha Cabral 14, 1250-012 Lisbon, Portugal; 3Department of Ophthalmology, Cuf Descobertas Hospital, Rua Mário Botas, 1998-018 Lisbon, Portugal; 4Clinical Analysis Laboratory Dr. Joaquim Chaves, Rua Aníbal Bettencourt, nº3, Edificio CORE, 2790-225 Carnaxide, Portugal; carlos.cardoso@jcs.pt (C.C.); rui.pinto@jcs.pt (R.P.); 5Center for the Study of Vision Sciences, University Ophthalmology Clinic, Faculty of Medicine, University of Lisbon, Av. Professor Egas Moniz, Piso 1C, 1649-028 Lisbon, Portugal; 6Department of Gynecology, Obstetrics and Pediatrics, Faculty of Medicine, University of Porto, Alameda Prof. Hernâni Monteiro, 4200-319 Porto, Portugal; hguimara@med.up.pt

**Keywords:** retinopathy of prematurity, neonatal diseases, fetal hemoglobin, pathophysiology, innovative therapeutic approaches, neonatal anemia, red blood cell transfusions, preterm infants

## Abstract

**Background/Objectives:** Retinopathy of prematurity (ROP) is a leading cause of vision impairment in preterm infants, with its pathogenesis linked to oxygen exposure. Red blood cell (RBC) transfusions, commonly performed in neonatal intensive care units (NICUs), reduce fetal hemoglobin (HbF) fraction, altering oxygen dynamics and potentially contributing to ROP. We aimed to investigate the relationship between RBC transfusions, HbF percentage, and ROP, evaluating HbF as a potential predictive biomarker. **Methods:** A multicenter, prospective study was conducted across eight Portuguese NICUs, involving infants born at <32 weeks gestational age (GA) or <1500 g. ROP staging followed the International Classification of ROP (ICROP2). Clinical data were collected during hospitalization, and HbF fractions were measured from blood samples in the first four weeks of life using standardized methods. Infants were stratified by ROP presence and treatment requirement. Statistical analysis was performed using SPSS 28.0, with *p* < 0.05. **Results:** Eighty-two infants (mean GA: 28.1 ± 2.1 weeks, birth weight: 1055.8 ± 258.3 g) were included. Among them, 29 (35.4%) presented ROP and 4 (4.9%) required treatment. Infants with ROP had more RBC transfusions and lower HbF percentages than those without ROP (*p* < 0.05). Lower HbF was associated with more RBC transfusions (*p* < 0.001). Kaplan–Meier survival curves showed a higher ROP risk in infants with reduced HbF (*p* < 0.05). **Conclusions:** Low HbF percentage in the first four weeks of life may increase ROP risk in preterm infants. HbF could serve as a biomarker for ROP prediction. Interventions preserving HbF may reduce ROP risk. Further studies are needed to validate HbF as a biomarker and refine prevention strategies.

## 1. Introduction

The birth of an extremely preterm infant poses considerable challenges. While advancements in neonatal care have significantly improved survival rates, concerns about long-term morbidities and associated functional impairments remain, with these factors profoundly affecting the well-being and quality of life of preterm infants [1]. Among the many complications of prematurity, retinopathy of prematurity (ROP) stands out as a major cause of visual impairment and childhood blindness worldwide [2].

Globally, the incidence of ROP varies widely, ranging from approximately 20 to 30% among preterm infants in high-income countries to significantly higher levels in low- and middle-income regions, where access to advanced neonatal care is limited [3]. ROP arises from the abnormal development of retinal blood vessels in preterm infants, driven by fluctuating oxygen levels after birth. The pathogenesis of ROP unfolds in two distinct phases: an initial phase marked by high oxygen exposure, which halts normal retinal vascularization, followed by a second phase characterized by retinal ischemia and abnormal neovascularization. This neovascularization is mediated by hypoxia-induced angiogenic factors, such as vascular endothelial growth factor (VEGF) and erythropoietin [4,5]. If untreated, these pathological changes can progress to retinal detachment and vision loss.

ROP is classified according to the International Classification of Retinopathy of Prematurity (ICROP), now in its third edition (ICROP3) [6,7,8]. This system defines the disease’s severity using five stages:
Stage 1: A thin demarcation line between vascularized and avascular retina.Stage 2: A ridge forms at this demarcation line.Stage 3: Abnormal neovascularization extends into the vitreous.Stage 4: Partial retinal detachment.Stage 5: Total retinal detachment, potentially leading to blindness.

Additionally, treatment guidelines are based on evidence from the Early Treatment for Retinopathy of Prematurity (ET-ROP) Study, which established criteria for type 1 and type 2 ROP [9]. These criteria incorporate the zones of the retina, which describe the location of disease relative to the optic disc: zone I is a circle centered on the optic disc with a radius twice the distance to the macula (the area closest to the disc and most critical); zone II extends from the edge of zone I to the nasal ora serrata; and zone III is the remaining peripheral crescent of the retina. The presence of plus disease, characterized by increased vascular dilation and tortuosity, is also a key factor.

Type 1 ROP, requiring prompt treatment, is defined as disease in zone I (any stage with plus disease or stage 3 without plus disease) or zone II (stage 2 or 3 with plus disease). Type 2 ROP, suitable for close monitoring, includes less severe disease in zone I (stage 1 or 2 without plus disease) or zone II (stage 3 without plus disease) [9]. This classification aids in clinical decision-making and ensures timely intervention when necessary.

Despite improved neonatal care and preventive measures, such as optimized oxygen therapy protocols, ROP remains a significant clinical challenge [10]. This highlights the urgent need for innovative strategies to mitigate its impact.

Emerging evidence, including our previous studies, suggests that maintaining higher percentages of fetal hemoglobin (HbF) during the early extra-uterine life of preterm infants may reduce the risk of ROP [11,12]. Fetal red blood cells (RBC) contain a high proportion of HbF, which has a greater oxygen affinity than adult hemoglobin (HbA) [13]. During the third trimester of pregnancy, HbF accounts for 70–80% of total hemoglobin. Between 32 and 36 weeks of gestational age (GA), there is a significant transition to HbA. Subsequently, the HbF fraction gradually decreases to 3–4% by around 24 weeks post-term age [14,15]. Interestingly, the period of HbF reduction between 32 and 36 weeks of GA coincides with the onset of ROP, suggesting a possible link between HbF percentage and the risk of developing this condition.

Although primarily genetically determined, HbF percentages can also increase in response to hypoxemia, cardiopulmonary insufficiency, or severe anemia in preterm infants. Maternal conditions, such as hypoxia, diabetes, or intrauterine growth restriction, may delay the transition from HbF to HbA [16]. Conversely, adult RBC transfusions, commonly administered to preterm infants during the neonatal period, reduce the HbF fraction, replacing it with HbA. This replacement may contribute to ROP by increasing dissolved oxygen in plasma, exposing the retina to excessive oxygen after a 15 mL/kg RBC transfusion [12]. This elevation induces a rightward shift in the oxygen dissociation curve, leading to an exaggerated, non-physiological supply of oxygen to retinal tissue during a critical phase of susceptibility to hyperoxia, thereby decreasing the angiogenic drive [17,18,19]. Additionally, the immature antioxidant systems of preterm infants exacerbate oxidative stress following RBC transfusions [20], while inflammatory responses and endothelial activation further contribute to ROP pathogenesis [21]. Notably, preterm infants—especially those critically ill—constitute one of the highest transfusion-requiring cohorts [22].

Given the pivotal role of HbF in oxygen transport and its potential impact on retinal vascularization, this study aims to evaluate HbF as a predictive biomarker for ROP. By exploring HbF’s potential to mitigate ROP risk, we hope to enhance the understanding of the disease’s mechanisms and improve risk stratification in preterm infants. The broader goal is to advance our comprehension of predictive and preventive measures.

## 2. Materials and Methods

### 2.1. Study Design and Population

This multicenter, observational, and analytical study involved a cohort of preterm infants from eight neonatal intensive care units (NICUs) in Portugal, namely Centro Hospitalar Universitário de São João, Centro Hospitalar Universitário de Lisboa Norte, Hospital Prof. Doutor Fernando Fonseca, Centro Materno Infantil do Norte belonging to the Centro Hospitalar Universitário do Porto, Hospital da Senhora da Oliveira, Guimarães, Hospital de Braga, and Maternidade Daniel de Matos and Maternidade Bissaya Barreto belonging to the Centro Hospitalar Universitário de Coimbra. The study is registered as an observational study (ISRCTN16889608).

Recruitment occurred between 19 November 2018, and 19 December 2019, with the study concluding between 30 December 2020, and 21 July 2021. The study followed preterm infants consecutively from birth, including both sexes and all races, if they met at least one of the following inclusion criteria: (1) born before 32 weeks of GA, or (2) birth weight (BW) below 1500 g. Exclusion criteria were as follows: (1) major congenital malformations, (2) congenital or acquired ophthalmological conditions (excluding conjunctivitis, keratitis, and congenital nasolacrimal duct obstruction) unrelated to ROP within the first 12 weeks of life, (3) death prior to the first ROP screening, (4) incomplete clinical data due to patient transfer to another hospital, and (5) lack of informed consent from parents or legal guardians.

The initial cohort for this study consisted of 455 preterm infants enrolled in the GenE-ROP Study, conducted in the eight NICUs mentioned. For the present analysis, a subgroup of 82 infants was selected based on the availability of sufficient blood samples to enable a consistent longitudinal assessment of HbF percentages during the first four weeks of life.

The detailed methodology of the GenE-ROP Study has been described in our earlier publications [23,24].

### 2.2. Screening for ROP and Gathering Ophthalmological Data

The initial retinal screening was conducted at 31–33 weeks postmenstrual age (PMA) or 4–6 weeks after birth, whichever was later, by qualified ophthalmologists at each NICU. The screening process adhered to the following procedures: (1) adequate mydriasis was achieved using eye drops, following the Portuguese Society of Neonatology’s consensus guidelines on ROP [25]; (2) screenings were conducted with either indirect binocular ophthalmoscopy, digital fundus retinography (RetCam), or a combination of both, depending on the specific indication and the protocols of each hospital center; (3) findings were documented for each eye based on the second edition of the ICROP (ICROP2). Follow-up screenings were scheduled according to ROP presence, zone, and severity and were continued until full retinal vascularization or ROP remission post-treatment. Data on the highest stage of ROP, any required treatment, and follow-up examinations were recorded per ICROP2 guidelines [6,7,26]. Treatment criteria followed those from the Early Treatment for Retinopathy of Prematurity (ETROP) Study [9]. Infants diagnosed with type 1 ROP were treated with anti-VEGF or LASER photocoagulation, depending on the disease location and the hospital center.

### 2.3. Other Data

Data regarding demographics and clinical conditions were gathered from the hospital’s medical records during the infants’ hospitalization, including GA, BW, sex, and clinical outcomes such as bronchopulmonary dysplasia, peri-intraventricular hemorrhage, cystic periventricular leukomalacia, necrotizing enterocolitis, and hemodynamically significant patent ductus arteriosus.

The analysis of HbF was conducted using the residual blood remaining after routine clinical tests performed on the neonates during their first four weeks of life. All biological samples were coded, anonymized, and stored at −20 °C to ensure patient confidentiality and safeguard their identities.

Hemoglobin analysis was performed using capillary electrophoresis (Capillarys™ Neonat Fast system, Sebia, Lisses, France). Capillary electrophoresis measures the HbF fraction by combining electrophoretic mobility in an alkaline buffer and electro-osmotic flow. A high voltage applied to a silica glass capillary causes hemoglobin to migrate toward a 415 nm wavelength detector. Hemoglobin fractions are quantified and displayed in a pherogram. Automated integration and defined migration zones (N1 to N13) assist in interpreting hemoglobin patterns.

Capillary electrophoresis detects and quantifies HbF, HbA, HbA2, HbS, HbC, HbDPunjab, HbOArab, HbE, Hb Lepore, and Hb Bart’s. It also identifies various Hb variants, including γ- and α-chain abnormalities. Internal quality controls, provided by Sebia (France), contain HbF, HbA, HbS, and HbC. The detection limit for HbA and HbS is approximately 1% of total hemoglobin.

### 2.4. Statistical Analysis

Statistical analysis was performed using the Statistical Package for Social Sciences (IBM^®^ SPSS^®^ Statistics, Chicago, IL, USA) version 28.0 for Windows^®^ Software. Categorical variables were expressed as *n* (%) and continuous variables as mean ± standard deviation or median (interquartile range). The χ^2^ tested differences between the groups for discrete variables, and *t*-test (parametric test) or Mann–Whitney/Kruskal–Wallis tests (non-parametric tests) for continuous variables, as appropriate. An adjustment was made using binary logistic regression. Survival analysis was conducted using the Kaplan–Meier method and differences between groups were assessed using the log-rank test. Statistical significance was defined as a *p*-value < 0.05.

In the original cohort of 455 preterm infants, a univariate logistic regression analysis was conducted to evaluate multiple risk factors for ROP, as previously described [23]. Variables identified as significant in the univariate analysis were included in a multivariate logistic regression model, which determined that GA and the number of RBC transfusions were the most statistically significant factors associated with ROP. These findings guided the adjustments performed in the current analysis.

### 2.5. Ethics Approval

All procedures involving human participants in this study adhered to the ethical standards set forth by the Ethics Committee of Centro Hospitalar Universitário de Lisboa Norte, the Centro Académico de Medicina de Lisboa (CAML), and the Ethics Committees of all other participating hospital centers. These procedures were also conducted in accordance with the 1964 Helsinki Declaration and its subsequent amendments, as well as comparable ethical standards. Written informed consent was obtained from the parents of all infants included in the study.

## 3. Results

This study included 82 preterm infants, with a mean GA of 28.1 ± 2.1 weeks and a mean BW of 1055.8 ± 258.3 g. Among them, 29 (35.4%) presented ROP, with four (4.9%) requiring treatment: two (2.4%) underwent laser photocoagulation, and two (2.4%) received anti-VEGF therapy. The cohort comprised 41 females (50%). Detailed clinical characteristics, including ROP stage distribution and associated prematurity-related comorbidities, are provided in Table 1.

Preterm infants with ROP received significantly more RBC transfusions during the first 4 weeks of life compared to those without ROP (Table 2).

A significant decrease in HbF percentages during the first four weeks of life was observed with increased RBC transfusions. Infants requiring more transfusions had notably lower HbF percentages than those needing fewer or none (*p* < 0.001) (Table 3).

Table 4 shows the association between HbF percentages during the first four weeks of life and the presence of ROP. Preterm infants who developed ROP had significantly lower mean HbF percentages during this period compared to those without ROP, even after adjusting for GA. Weekly data analysis revealed that mean HbF percentages were significantly and independently lower in infants with ROP from the second to the fourth week of life.

Additionally, preterm infants who required treatment for ROP (type 1 ROP) had significantly lower mean HbF percentages during the first four weeks of life compared to those with ROP who did not require treatment, even after adjusting for GA (Table 5). When assessed weekly, however, this association was significant only during the second and third weeks of life and was not independent of GA during these periods.

Figure 1 shows Kaplan–Meier survival curves for ROP diagnosis in preterm infants categorized into quartiles for HbF percentage for the first 4 weeks of life and each week within that timeframe at specific PMA and postnatal ages. These survival curves support our previous findings, indicating that groups of preterm infants with lower HbF percentages in the first four weeks of extra-uterine life were more likely to develop ROP than groups with higher HbF percentages. These results remain statistically significant in the HbF groups from the second to fourth week of life.

## 4. Discussion

RBC transfusions are essential for the survival of prematurely born infants experiencing severe anemia or hemorrhage. However, these transfusions may also contribute to common morbidities observed in NICU settings.

A notable association has been established between RBC transfusions administered during the first four weeks of life and the subsequent development of ROP in our cohort. This association remained significant even after accounting for GA. Previous studies support this link. For example, Lust et al. reported a fourfold increased risk of severe ROP with transfusions during the first ten days of life [19]. Similarly, Teofili et al. found that infants with severe ROP received transfusions earlier, as measured by PMA or postnatal days [22]. Del Vecchio et al. observed that reducing RBC transfusions decreased the incidence of ROP, bronchopulmonary dysplasia, and necrotizing enterocolitis [18].

Emerging evidence suggests that RBC transfusions elevate free iron levels and oxidative stress in preterm infants, contributing to ferroptosis, an iron-dependent form of programmed cell death [27,28]. Our prior research links high erythroblast counts during the first week of extra-uterine life with ROP development, suggesting that anemia and transfusions may play roles in this process [23]. Increased erythroblast production during anemia results in erythroferrone (ERFE) secretion, which suppresses hepatic hepcidin synthesis, leading to enhanced iron absorption [29]. This iron overload, compounded by free iron from transfusions, raises the risk of retinal ferroptosis, potentially damaging the blood–retina barrier and contributing to ROP pathogenesis [28]. Ferroptosis, therefore, may represent a mechanistic link between anemia, transfusions, and ROP, warranting further investigation.

Approximately 50% of infants born before 32 weeks of GA and 85% of those weighing less than 1000 g receive RBC transfusions during their NICU stay [13].

Neonatal anemia is defined as hemoglobin or hematocrit levels below the average, typically more than two standard deviations below the mean for the infant’s age after birth [30]. Except for extremely preterm infants, hemoglobin levels at birth are higher than those later in life, compensating for relative intrauterine hypoxia [13]. After birth, hemoglobin levels decrease as the rate of RBC death exceeds the replacement rate [13]. Physiological anemia is common in nearly all newborns during the first months of life and is more pronounced in very preterm infants, primarily due to their impaired ability to increase erythropoietin production in immature kidneys [13,31]. Other significant causes of neonatal anemia include phlebotomy-related blood losses for laboratory testing and increased RBC degradation [32]. Additionally, in sick preterm infants, hematocrit levels decline further after birth due to various other factors. It has been shown that low hematocrit levels adversely affect short-term outcomes in preterm infants [33].

Given the apparent association between low HbF percentages in early life and the development of ROP in preterm infants, we aimed to evaluate HbF percentages during the first four weeks of life as a potential predictive biomarker for ROP development. Our study found that lower mean HbF percentages during this period were strongly predictive of ROP. In particular, the decline in HbF percentages, especially between the second and fourth week of life, was correlated with an increased susceptibility to ROP. This finding aligns with the cumulative effect of adult RBC transfusions in reducing the HbF fraction. Furthermore, a study by Stutchfield et al. reported that preterm infants with lower HbF fractions during their NICU stay were at a higher risk for ROP development [17]. Similarly, Hellström et al. found that low HbF fractions during the first week of extra-uterine life independently predicted ROP in preterm infants [14]. A study by Teofili et al. also revealed that preterm infants receiving RBC transfusions from adult donors exhibited a 40% decrease in HbF percentages with each transfusion [34]. Notably, these percentages recovered in the absence of further transfusions; however, repeated transfusions resulted in an irreversible decline in HbF levels, underscoring the critical importance of the timing of successive transfusions [34].

In our analysis, lower mean HbF percentages in the first four weeks of life were associated with developing severe ROP requiring treatment. Weekly analysis showed that this correlation remained significant during the second and third weeks of life, but it was not independent of GA. It is worth noting that the treatment-requiring group was small, consisting of only four patients, which may have influenced the results of this sub-analysis.

When adult blood is transfused to preterm infants, it replaces the physiological content of HbF with HbA. HbF differs from HbA because it has two ɣ subunits, while HbA has two β subunits. The ɣ subunit in HbF has a lower binding affinity for 2,3-diphosphoglycerate in the deoxygenated state, effectively increasing its oxygen binding affinity [15]. The Bohr effect offsets this potential disadvantage, where hemoglobin releases oxygen in response to increased CO_2_ and decreased pH [16]. These characteristics give HbF a higher oxygen affinity, a leftward shift in the oxyhemoglobin dissociation curve, and a steeper oxyhemoglobin dissociation curve. Consequently, HbF delivers oxygen to tissues more effectively by releasing oxygen earlier during hypoxia and with minimal changes in tissue oxygenation [35]. Additionally, HbF impedes the release of heme groups and demonstrates increased pseudo-peroxidase activity, facilitating the rapid reconversion of reactive ferryl heme. Furthermore, HbF has a greater capacity to produce unbound nitric oxide through oxidative denitrosylation, which regulates vascular tone and blood flow [14,22]. The PacIFiHER study highlighted that low HbF fractions in preterm infants are associated with an increased risk of ROP and poorer systemic oxygenation indices [16].

It is well established that even a small increase in the HbF fraction can significantly benefit the management of sickle cell disease and β-thalassemia. In β-thalassemia, this benefit is linked to HbF’s potent antioxidant capacity, which helps reduce the oxidative stress that contributes to hemolytic crises. Additionally, HbF’s higher oxygen-binding affinity compared to HbA allows it to capture more oxygen in the lungs and placenta. At the tissue level, its greater nitrite reductase activity promotes vasodilation and facilitates oxygen delivery. Together, these properties underscore HbF’s protective role in conditions like sickle cell disease and β-thalassemia, as well as its essential function in supporting preterm infants’ transition to an oxygen-rich environment after birth [22].

Therefore, HbF’s high oxygen affinity and unique structure may provide significant protective benefits across the two phases of ROP development. In the initial hyperoxic phase, where postnatal oxygen levels inhibit retinal vascularization, HbF may support controlled oxygen delivery, alleviating oxidative stress and reducing the risk of retinal damage. In the subsequent hypoxic phase, an elevated HbF fraction may enhance oxygen transport to retinal tissues, helping to counterbalance the low-oxygen environment and mitigate hypoxia-driven damage. This dual function of HbF underscores its potential protective role against the oxygen fluctuations that contribute to ROP development, suggesting that a lower HbF fraction in early life may serve as both a biomarker and a modifiable risk factor, reinforcing the value of HbF-sparing strategies in mitigating ROP progression.

The current treatments for ROP primarily involve anti-VEGF therapy and laser photocoagulation, which target proliferative ROP. However, it is essential to explore additional interventions focused on preventing ROP. Our research and previous studies highlight the importance of identifying high-risk infants and enhancing their natural protective mechanisms, such as avoiding the rapid decrease in HbF early in life, to minimize or prevent ROP development. Strategies to reduce the use of adult RBC transfusions could be instrumental in achieving this goal. Potential approaches include implementing delayed umbilical cord clamping, adopting blood-sparing sampling techniques to minimize blood loss, and considering the use of placental umbilical cord RBC instead of adult RBC for transfusions in preterm infants [12,14,36]. Another potential alternative is the administration of hydroxyurea to elevate the HbF fraction in high-risk infants [35].

Delayed cord clamping at birth has proven highly effective in preventing neonatal anemia [14]. According to a Cochrane meta-analysis, the resultant autotransfusion of fetoplacental blood reduces the need for RBC transfusions [37], which helps maintain HbF levels [17].

A multicenter study is currently evaluating the impact of microsampling on HbF preservation to reduce neonatal morbidities, including ROP (NCT04239690) [14].

Using umbilical cord blood, rather than adult blood, for RBC transfusions in preterm infants is another promising strategy for preventing complications related to low HbF percentages caused by adult RBC transfusions [38]. Umbilical cord blood, typically discarded after delivery, contains nearly 100% HbF. According to Podraza, approximately 100 mL of blood can be collected from the umbilical cord after the delivery of a full-term newborn. For a preterm infant weighing 500 g, only 7.5 mL of blood is needed for transfusion, based on the standard requirement of 15 mL/kg [12]. While umbilical cord blood transfusion represents an innovative approach to ROP prevention, it poses logistical challenges, including the establishment of cord blood banks, distribution systems, and considerations related to cost-effectiveness [12,39]. This approach, although complex, aims to administer RBC exclusively rich in HbF for all transfusions, thereby reducing the HbA content in the neonatal circulation. The ongoing multicenter randomized clinical trial BORN study is investigating the potential benefits of this method (NCT05100212) [40].

Future research should focus on assessing the efficacy and feasibility of implementing these strategies in clinical settings to determine their potential for managing ROP in preterm infants.

Current evidence suggests that stricter transfusion protocols may help mitigate transfusion-related complications in preterm infants [41,42]. A survey conducted in European NICUs revealed variability in transfusion practices, with many centers using more liberal thresholds compared to recent trials advocating for more restrictive practices [43]. Further investigation is required to understand the factors influencing the incorporation of these trial findings in neonatal transfusion protocols.

Our study has several limitations that need to be considered. Firstly, the relatively small sample size, particularly regarding the limited number of infants diagnosed with ROP requiring treatment, increases the risk of Type II error and limits the generalizability of our findings to broader populations. This limitation is inherent to the study design, as we focused on a subgroup of 82 preterm infants with sufficient blood samples for consistent longitudinal HbF assessment during the first four weeks of life. While this approach ensured robust analyses within a consistent cohort, it inherently restricted the scope of our findings. Additionally, our investigation focused solely on the occurrence of ROP and the presence of severe ROP requiring treatment rather than encompassing all stages of ROP, which may also limit the scope of our findings. Another limitation is that our investigation relied on blood samples collected during routine clinical care, without strictly standardized timing for sample collection. Furthermore, although we adjusted for GA, a key risk factor for ROP identified in prior analyses of the full cohort of 455 preterm infants [23], the complexity of ROP pathogenesis and the observational nature of the study limited our ability to fully account for all potential confounders. Despite these limitations, our findings demonstrated a robust association between HbF percentages in the initial four weeks of extra-uterine life and the development of ROP. This supports HbF’s potential as a predictive biomarker for ROP.

Future studies with larger, multicenter cohorts are needed to validate our findings, further investigate the role of HbF in ROP pathogenesis, and evaluate preventive strategies aimed at preserving HbF levels, such as reducing adult RBC transfusions or employing alternative transfusion approaches. Such research will be essential to advancing our understanding of ROP and improving outcomes for preterm infants.

To summarize the study’s findings and implications, Figure 2 presents a comprehensive flowchart highlighting the key aspects and take-home messages of the research.

## 5. Conclusions

Our study highlights the significant role of hemoglobin fractions, particularly HbF, in the pathogenesis of ROP among preterm infants. The consistent association between lower HbF percentages and increased risk of ROP underscores its potential as a valuable, low-cost predictive biomarker for identifying high-risk infants. Incorporating HbF monitoring into neonatal care could refine risk stratification and enhance early screening protocols, enabling more targeted interventions.

Additionally, the potential benefits of reducing adult RBC transfusions—through strategies such as delayed cord clamping, cord blood transfusions, and restrictive transfusion protocols—warrant further exploration. These interventions may help mitigate the harmful effects of oxidative stress and iron overload, both of which contribute to ROP pathogenesis.

While this study emphasizes HbF’s importance, the findings are limited by the relatively small sample size and observational design, necessitating further validation in larger cohorts. Such research will be crucial to confirming HbF’s role as a biomarker and evaluating the effectiveness of preventive strategies. Expanding our understanding of HbF’s protective role could ultimately improve outcomes for preterm infants at risk of ROP and reduce the global burden of this condition.

## Figures and Tables

**Figure 1 biomedicines-13-00110-f001:**
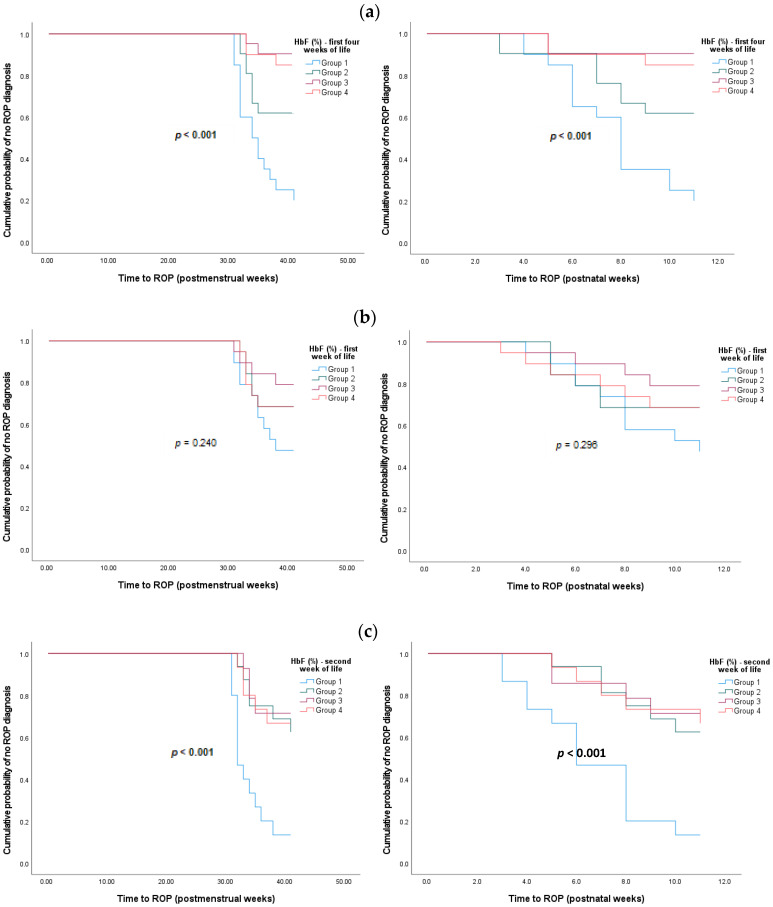
Kaplan–Meier survival curves for ROP diagnosis of preterm infants at specific PMA and postnatal ages, based on HbF percentages divided into quartiles for (**a**) mean of the first four weeks of life (*n* = 82); (**b**) the first week of life (*n* = 76); (**c**) the second week of life (*n* = 60); (**d**) the third week of life (*n* = 55); (**e**) the fourth week of life (*n* = 49). Group 1: minimum–first quartile; Group 2: first quartile–second quartile; Group 3: second quartile–third quartile; Group 4: third quartile–maximum. *p*-values less than 0.05 are in bold.

**Figure 2 biomedicines-13-00110-f002:**
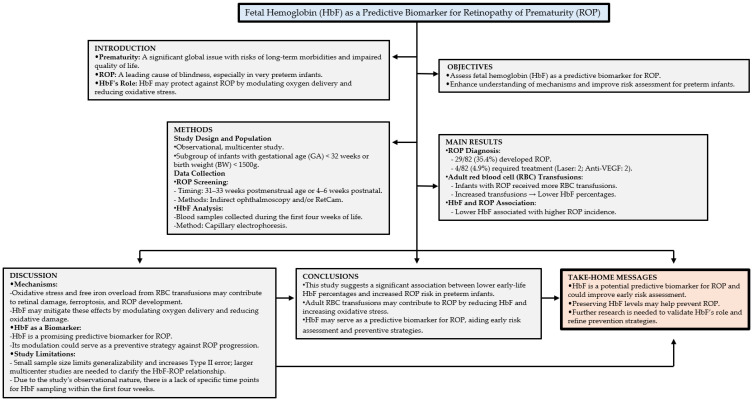
Study overview and key insights. Flowchart summarizing the study’s design, results, and practical implications of HbF for ROP.

**Table 1 biomedicines-13-00110-t001:** Demographic and clinical characteristics of preterm infants in the study population, including ROP stages.

Number of infants (*n*)	82
GA (weeks), mean ± SD	28.1 ± 2.1
BW (grams), mean ± SD	1055.8 ± 258.3
Female, *n* (%)	41 (50.0)
Male, *n* (%)	41 (50.0)
Retinopathy of prematurity, *n* (%)	29 (35.4)
Stage 1 ROP, *n* (%)	12 (14.6)
Stage 2 ROP, *n* (%)	9 (11.0)
Stage 3 ROP, *n* (%)	4 (4.9)
Type 1 ROP, *n* (%)	4 (4.9)
Bronchopulmonary dysplasia, *n* (%)	21 (25.6)
Peri-intraventricular hemorrhage, *n* (%)	12 (14.6)
Cystic periventricular leukomalacia, *n* (%)	3 (3.7)
Necrotizing enterocolitis, *n* (%)	14 (17.1)

*n*, number of individuals; SD, standard deviation.

**Table 2 biomedicines-13-00110-t002:** Comparison of RBC transfusions administered during the first 4 weeks of life between infants who developed ROP and those who did not.

Number of RBC Transfusions (First 4 Weeks)	No ROP (*n*, %)	ROP (*n*, %)	*p*	*p**
0	43 (81.1)	8 (27.6)	**<0.001 ^1^**	**0.009**
1–2	9 (17.0)	13 (44.8)		
3–4	1 (1.9)	8 (27.6)		

*p*, *p*-value; *p**, binary logistic regression, adjusted for gestational age (GA). ^1^ Chi-square test. *p*-values less than 0.05 are in bold.

**Table 3 biomedicines-13-00110-t003:** Association between the number of RBC transfusions and mean HbF percentages in the first 4 weeks of life.

Number of RBC Transfusions (First 4 Weeks)	Mean HbF (%)—First 4 Weeks *n*; Median (Q1–Q3)	*p*
0	51; 77.3 (75.2–81.8)	**<0.001 ^1^**
1–2	22; 67.8 (60.4–75.2)	
3–4	9; 44.5 (34.7–53.8)	

^1^ Kruskal–Wallis test. *p*-value in bold indicates statistical significance (*p* < 0.05).

**Table 4 biomedicines-13-00110-t004:** Association between HbF in the first four weeks of life and ROP.

HbF (%)	No ROP *n*; Median (Q1–Q3) or Mean ± SD	ROP *n*; Median (Q1–Q3) or Mean ± SD	*p*	*p**
Mean first 4 weeks	53; 77.3 (74.4–81.6)	29; 62.7 (51.0–72.8)	**<0.001 ^1^**	**0.003**
1st week of life	50; 79.0 (76.4–81.0)	26; 76.9 (69.2–80.6)	0.100 ^1^	0.693
2nd week of life	32; 77.8 (74.6–81.2)	28; 73.3 (54.4–79.5)	**0.005 ^1^**	**0.020**
3rd week of life	29; 76.1 (69.7–79.8)	26; 52.5 (33.7–73.4)	**<0.001 ^1^**	**0.031**
4th week of life	24; 70.5 ± 17.3	25; 44.9 ± 19.2	**<0.001 ^2^**	**0.019**

*p**, binary logistic regression, adjusted for GA; ^1^ Mann–Whitney test. ^2^ *t*-test. *p*-values less than 0.05 are in bold.

**Table 5 biomedicines-13-00110-t005:** Association between HbF in the first four weeks of life and ROP requiring treatment (type 1 ROP).

HbF (%)	No Type 1 ROP *n*; Median (Q1–Q3) or Mean ± SD	Type 1 ROP *n*; Median (Q1–Q3) or Mean ± SD	*p*	*p**
Mean first 4 weeks	25; 64.3 ± 13.3	4; 40.1 ± 7.8	**0.002 ^1^**	**0.046**
1st week of life	22; 77.4 (71.7–81.8)	4; 56.9 (37.3–77.7)	0.112 ^2^	0.119
2nd week of life	24; 75.1 (65.3–79.7)	4; 43.3 (30.8–58.3)	**0.005 ^2^**	0.057
3rd week of life	22; 59.5 ± 19.5	4; 23.8 ± 5.7	**<0.001 ^1^**	0.202
4th week of life	21; 46.7 ± 18.5	4; 35.3 ± 22.8	0.287 ^1^	0.507

*p**, binary logistic regression, adjusted for GA; ^1^ *t*-test. ^2^ Mann–Whitney test. *p*-values less than 0.05 are in bold.

## Data Availability

The datasets analyzed in this study are available from the corresponding author upon reasonable request.

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
