# Peer review of "Fetal Hemoglobin as a Predictive Biomarker for Retinopathy of Prematurity: A Prospective Multicenter Cohort Study in Portugal"

_biomedicines, 2025, doi:10.3390/biomedicines13010110_

Round 1

Reviewer 1 Report

Comments and Suggestions for Authors

The authors report on a multi-center observational study (eight participating neonatal intensive care units in Portugal) with the aim to assess fetal hemoglobin as a predictive biomarker for the development of retinopathy of prematurity. A total of 82 preterm infants are included in the study. They observe that higher percentages of fetal hemoglobin / fewer transfusions with adult blood, may have a role in reducing the risk of developing retinopathy of prematurity. The paper is well written and the data are clearly presented. Limitations of the study are discussed appropriately. I have only a minor comment:

Figure 1: Indicate HbF range for each of the quartiles; Y axis: label (Cumulative survival) is misleading, this refers to ROP diagnosis; indicate n (presumably 82).

Comments on the Quality of English Language

None.

Author Response

We sincerely thank you for your positive feedback on our study and your thoughtful suggestions, which have helped us improve the manuscript further. As suggested, we have carefully revised the English language throughout the manuscript to enhance clarity and readability. Below, we address your specific comment and outline the changes made in the revised version.

Answer:

Since the values for each quartile vary week by week, we decided to mention them as follows: 'Group 1: minimum–first quartile; Group 2: first quartile–second quartile; Group 3: second quartile–third quartile; Group 4: third quartile–maximum.' This approach avoids overloading the figure or its caption. We have included the quartile values below, and if you believe it is better, we can add them to the figure.

Regarding n, we chose not to include it directly in the figure. However, n corresponds to the sum of the ROP and non-ROP groups from Table 4, and it varies by week. This variability reflects the observational nature of the study, as we were unable to ensure blood samples from all infants in every week of the analysis. To enhance clarity, we have now added the n values to the figure caption as well, in case you prefer this approach.

As for the Y-axis, we have replaced "Cumulative survival" with "Cumulative probability of no ROP diagnosis" to make it clearer.

Thank you again for your valuable feedback, which has allowed us to make these improvements and enhance the overall clarity and presentation of the manuscript.

                                              Table – N and HbF range for each of the quartiles

Mean first 4 weeks

1st week of life

2nd week of life

3rd week of life

4th week of life

n=82

Group 1:

31.3%-65.0%

Group 2:

66.1%-75.3%

Group 3:

75.9%-79.4%

Group 4:

79.5%-97.7%

n=76

Group 1:

18.6%-75.1%

Group 2:

75.4%-78.0%

Group 3:

78.6%-80.9%

Group 4:

81.0%-97.7%

n=60

Group 1:

27.0%-71.7%

Group 2:

72.3%-76.8%

Group 3:

76.9%-79.9%

Group 4:

81.1%-96.6%

n=55

Group 1:

17.5%-47.2%

Group 2:

47.6%-70.6%

Group 3:

71.9%-78.5%

Group 4:

78.7%-94.5%

n=49

Group 1:

19.1%-39.8%

Group 2:

40.7%-60.3%

Group 3:

62.3%-75.3%

Group 4:

75.7%-95.3%

Group 1: minimum-first quartile; Group 2: first quartile-second quartile; Group 3: second quartile-third quartile; Group 4: third quartile-maximum.

Reviewer 2 Report

Comments and Suggestions for Authors

The manuscript titled "Fetal Hemoglobin as a Predictive Biomarker for Retinopathy of Prematurity: A Prospective Multicenter Cohort Study in Portugal" by Mariza Fevereiro-Martins et al.

- Title, the word "Portu-gal" should be continuous and not disrupted by hyphen. Should be Portugal.

- It will be nice if pictures/photos showing normal infants versus infants with retinopathy of prematurity.

- Table 1, it is unclear if only Female infants are studied? How about the data of the male infant counterparts?

- The manifestation of different stage of ROP should be defined clearly, to let the reader knows what're their differences.

- Also, it may be confusing to the reader for Type 1 ROP versus Stage 1 ROP. All these should be elaborated and explained better.

- Finally, a flow chart summarizing the essence of the current study and take home message should be provided, to enhance the knowledge transfer.

Comments on the Quality of English Language

Typos and unfriendly mode of English usage can be found.

Author Response

Thank you for your valuable comments and suggestions, which have significantly contributed to improving the quality of our manuscript. As suggested, we have carefully revised the English language throughout the manuscript to ensure clarity and precision. Below, we address each of your comments in detail and outline the changes made in the revised manuscript.

  • Title, the word "Portu-gal" should be continuous and not disrupted by hyphen. Should be Portugal.

Answer:

Thank you for your observation. We confirm that the word "Portugal" was corrected to appear continuous, without a hyphen, in the title of the version we initially submitted. After identifying this issue, we reviewed the manuscript again to ensure the formatting was accurate. While we are confident that the Word version in this submission has been properly formatted, we also submitted the PDF version of the revised manuscript that safeguards the formatting. We hope this resolves the issue, and we appreciate your attention to detail.

  • It will be nice if pictures/photos showing normal infants versus infants with retinopathy of prematurity.

Answer:

Thank you for your thoughtful suggestion. However, we were unable to include images or photographs showing normal infants versus infants with ROP, as these images were not collected as part of the study. More importantly, informed consent for using such photographs was not obtained from the parents of the infants involved, which prevented us from including them in the manuscript. We appreciate your understanding on this matter.

  • Table 1, it is unclear if only Female infants are studied? How about the data of the male infant counterparts?

Answer:

Thank you for your valuable feedback. To address your question, we would like to clarify that both male and female infants are included in the study. In the revised Table 1, we have explicitly listed the number of male infants (41) alongside the female infants (41), with each group representing 50% of the cohort. This was based on the actual distribution of the study population. Initially, we indicated only the number of female infants and their corresponding percentage (50%), assuming that the distribution of the remaining cohort as male would be implicit. However, we understand the importance of ensuring full clarity and have now updated the table accordingly. We hope this revision resolves any potential ambiguity.

  • The manifestation of different stage of ROP should be defined clearly, to let the reader knows what're their differences.

Answer:

Thank you for your thoughtful feedback. We have addressed this concern by elaborating on the description of the stages of ROP in the introduction. Specifically, we have now included a detailed explanation of the five stages of ROP, outlining the progressive retinal changes for each stage (lines 62 to 70 in the revised manuscript). These updates provide a clearer understanding of the differences between the stages, ensuring that readers can easily follow the disease progression. Since the description in lines 167–170 of the previous manuscript covered only part of this information, we removed it to avoid repetition and ensure consistency throughout the text.

  • Also, it may be confusing to the reader for Type 1 ROP versus Stage 1 ROP. All these should be elaborated and explained better.

Answer:

Thank you for raising this important point. Part of this concern has been addressed in the previous response detailing the stages of ROP. Additionally, we have clarified the distinction between Type 1 and Stage 1 ROP in the introduction (lines 71 to 83 in the revised manuscript). Type 1 and Type 2 ROP are clinical categories used to guide treatment decisions, while the stages describe the progressive changes in retinal pathology. This distinction has been explicitly elaborated to avoid any confusion for readers.

  • Finally, a flow chart summarizing the essence of the current study and take home message should be provided, to enhance the knowledge transfer.

Answer:

We sincerely thank you for suggesting the inclusion of a flowchart summarizing the essence of our study and the take-home messages. This addition has significantly enhanced our manuscript by providing a clear and concise visual representation of our study's design, key findings, mechanisms, conclusions, and practical implications. In the revised manuscript, we have referenced the flowchart at the end of the Discussion section, just before the Conclusions (lines 483 to 485), as we believe this placement effectively summarizes the key points discussed and provides a logical transition to the final remarks. If this placement or format differs from your vision, we are more than willing to adjust it to better align with your expectations. Your valuable input has greatly strengthened our manuscript, and we truly appreciate your thoughtful feedback.

Reviewer 3 Report

Comments and Suggestions for Authors

File attached

Author Response

We sincerely thank you for your thoughtful and constructive comments, which have helped us improve the manuscript further. Following your suggestions, we have carefully revised the manuscript, including a thorough review of the English language to enhance clarity and readability. Below, we address each of your points individually and outline the corresponding changes made in the revised version.

  1. the sample size of 82 infants limit the generalizability of the findings, the statistical power and potential biases need to be mentioned in methods.

Answer:

We sincerely appreciate your insightful comments regarding the potential limitations of our study, particularly concerning the sample size and potential biases. Due to budgetary constraints and the observational nature of the study, it was not possible to include the entire cohort of 455 preterm infants from the original GenE-ROP Study. Blood samples were not specifically collected for this study, and most infants in the larger cohort had only a limited number of samples available. To address these challenges, we focused our analysis on a subgroup of 82 infants with blood samples collected across most of the first four weeks of life. This selection allowed us to perform a robust longitudinal assessment of HbF levels within a consistent cohort of infants, rather than comparing data from different groups at different weeks.

In the original cohort of 455 preterm infants, univariate and multivariate logistic regression analyses identified gestational age (GA) and the number of days of red blood cell transfusions as the most significant risk factors for ROP, as detailed in a prior manuscript. Based on these findings, we adjusted for GA in our analysis, as it was the most significant risk factor aside from transfusions. Since HbF levels are directly influenced by transfusions, adjusting for transfusions was not appropriate.

We have revised the manuscript to clarify this rationale, and these changes are reflected in lines 150 to 154 and 209 to 214 of the revised manuscript. We are aware that smaller sample sizes can introduce potential biases, and other factors may influence outcomes. Nonetheless, the choice to adjust for GA reflects its established significance in the larger cohort and supports the validity of our findings.

These limitations, as well as the constraints imposed by the observational design, are explicitly discussed in the manuscript (lines 453 to 460 and 463 to 471). We trust that this explanation provides clarity regarding our analytical approach and demonstrates the rigor with which we sought to address potential biases.

  1. As per you, more HbF may reduce ROP risk. Have you explored about erythropoietin therapy, delayed cord clamping, or minimizing transfusions had role in preventing ROP?

Answer:

Thank you for your thoughtful comment and for raising these important points. We appreciate the opportunity to address them.

- Erythropoietin Therapy: We have information on which infants in our study received erythropoietin and which did not. Our analysis did not reveal any significant differences in the development of ROP between these two groups. However, we did not explore the relationship between erythropoietin administration and HbF levels. As our study was observational, HbF measurements during the first four weeks of life depended on available blood samples, with no standardized timing for collection. This limited our ability to assess the temporal relationship between erythropoietin administration and HbF levels.

- Delayed Cord Clamping: Unfortunately, we do not have data on whether delayed cord clamping was performed for any of the infants in our cohort. As a result, we were unable to explore its potential impact on ROP development.

- Minimizing Blood Transfusions: Our study did not assess the potential protective effect of minimizing red blood cell transfusions on ROP risk. Since this was an observational study without any medical intervention, there were no changes to standard neonatal care procedures in the NICUs where the study was conducted.

We are grateful for your valuable insights and hope these clarifications address your concerns.

  1. how does this study differ from prior findings [citations 19,20], particularly to increase HbF levels and reduce ROP risk?

Answer:

Thank you for your insightful question. We appreciate the opportunity to clarify this point. The main difference between the present study and our prior studies (doi:10.1007/s00417-023-06072-7 and doi:10.3390/children11101154) is that, in those earlier studies, we did not measure HbF levels. However, it was the findings from our prior research, particularly from doi:10.3390/ijms241411817 (reference 11 in our manuscript), that highlighted the potential importance of HbF in the context of ROP risk. This recognition informed the design of the current study, where we explicitly measured HbF levels and analyzed their relationship with the risk of ROP. This approach represents an advancement in our current study. We are grateful for your thoughtful feedback and hope this clarification is helpful.

  1. As evidenced in a Cochrane meta-analysis, the resultant autotransfusion of fetoplacental blood reduces the need for RBC transfusions [36], which helps maintain HbF. Is this study evaluated the ROP in their cohort?

Answer:

Thank you for your insightful question. We appreciate the opportunity to provide clarification. This Cochrane meta-analysis highlights that autotransfusion of fetoplacental blood, achieved through delayed cord clamping, can reduce the need for RBC transfusions, which may help maintain HbF levels. However, in our study, we did not specifically evaluate the impact of delayed cord clamping on the risk of ROP. Additionally, we do not have data on whether delayed cord clamping was performed for the infants in our cohort. As such, we were unable to assess the potential association between delayed cord clamping, HbF levels, and ROP risk.

With regard to the Cochrane meta-analysis, it is important to note that this review did not directly evaluate the relationship between delayed cord clamping and ROP incidence in the included studies. Its primary outcomes focused on neonatal and maternal health indicators, such as neonatal mortality, need for transfusions, and the risk of intraventricular hemorrhage.

We greatly appreciate your thoughtful comments.

  1. based on gestational age or birth weight subgroups is HbF level make any difference?

Answer:

Thank you for your insightful question regarding the impact of HbF levels across gestational age (GA) or birth weight (BW) subgroups. As previously noted, while BW may play a role among the multiple risk factors for ROP, our logistic regression analysis in the full cohort (455 preterm infants) identified GA and the number of red blood cell transfusions as the most significant independent risk factors, with BW losing significance in the multivariate model.

Although this analysis was not initially performed, we have now evaluated HbF levels across different GA and BW subgroups. Our findings indicate that infants with higher GA and BW had higher mean HbF levels. Notably, these groups also received significantly fewer RBC transfusions, which could explain the observed differences in HbF levels, as transfusions are known to significantly reduce HbF.

After adjusting for both GA and BW as covariates, the association between HbF levels and ROP development remained significant for some results, with the strongest significance observed for the mean HbF levels of the first four weeks of life, followed by the second week. Additionally, the relationship between the number of RBC transfusions and ROP remained significant and independent.

These findings reinforce the importance of HbF as a potential biomarker for ROP, demonstrating that even when accounting for these key risk factors, the relationship between HbF and ROP persists. Together, these results underline the robustness of our study and highlight the need for larger studies to confirm and expand upon these observations, as discussed in the manuscript.

  1. How this paragraph is relevant to ROP [It is well established that even a small increase in the HbF fraction can significantly benefit the management of sickle cell disease and β-thalassemia. These effects underscore HbF’s protective role against vaso-occlusive crises in sickle cell disease and its essential function in helping preterm infants transition to an oxygen-rich environment after birth [18].

Answer:

Thank you for your thoughtful feedback and for highlighting the need for further clarification. We appreciate the opportunity to address this point.

The relevance of this paragraph to ROP lies in the antioxidant and vasoregulatory properties of HbF. In β-thalassemia, increased HbF levels are beneficial due to its potent antioxidant capacity, which helps to counteract the oxidative stress that drives hemolytic crises in this condition. HbF also has a higher oxygen-binding affinity compared to adult hemoglobin (HbA), allowing it to capture more oxygen in the lungs and placenta. At the tissue level, HbF promotes vasodilation due to its enhanced nitrite reductase activity, facilitating improved oxygen delivery to tissues. This dual action of antioxidant protection and improved oxygen transport is relevant to preterm infants as it may help them transition more effectively to an oxygen-rich environment after birth.  This, in turn, may mitigate oxidative stress, which is a known contributor to ROP pathophysiology.

To improve clarity in the manuscript, we have revised the paragraph to better emphasize its relevance to ROP. The updated text can now be found in lines 383 to 392 of the revised manuscript.

We hope this revised explanation provides better context for the paragraph's relevance to ROP. Thank you once again for your valuable feedback. 

Round 2

Reviewer 2 Report

Comments and Suggestions for Authors

The authors have addressed to most of my concerns. Thank you!

Reviewer 3 Report

Comments and Suggestions for Authors

much improved and addressed my points